# Efficient Detection of Robot Kidnapping in Range Finder-Based Indoor Localization Using Quasi-Standardized 2D Dynamic Time Warping

Zool H. Ismail [1],* and Iksan Bukhori [2]

1    Centre for Artificial Intelligence and Robotics, Universiti Teknologi Malaysia, Jalan Sultan Yahya Petra, Kuala Lumpur 54100, Malaysia

2    Department of Electrical Engineering, Faculty of Engineering, President University, Bekasi 17550, Indonesia; iksan.bukhori@president.ac.id

\*    Correspondence: zool@utm.my; Tel.: +60-199816001

**Featured Application: In this work, an augmented online approach is used to effectively detect a kidnapping event within range-finder-based indoor localization.**

**Abstract:** This paper proposes an augmented online approach to detect kidnapping events within range-finder-based indoor localization. The method is specifically designed for an Internet of Things (IoT)-Aided Robotics Platform that enables the system to detect kidnapping across all time instances of an indoor mobile robotic operation with high accuracy and maintain a high accuracy in the face of relocalization failures. The approach is based on similarity degree of geometry shape of the environment obtained from range scan data between two consecutive time instances. The proposed approach named Quasi-Standardized Two-Dimensional Dynamic Time Warping (QS-2DDTW) is based on the Multidimensional Dynamic Time Warping (MD-DTW) with homogeneity variance test imbued in it. A series of simulations are preformed against maximum current weight, measurement entropy, and the four metrics in metric based detector. The result shows that the proposed method yields high performance in terms of its ability to distinguish kidnapping condition from normal condition and that it has low dependency towards relocalization process, thus ensures the accuracy of detection is not disturbed by relocalization.

**Keywords:** dynamic time warping; internet of things-aided robotic; indoor localization; kidnapping robot problem; variance homogeneity test

## 1. Introduction

Research trends within technological innovation are leading to the appearance of the Internet of Things (IoT)-aided robotics application. An IoT-aided robotic comprising an IoT device connected to a mobile platform plays a major role in tomorrow's society, continuing to help persons in accomplishing many duties, spanning from precision farming to industrial warehouses, from wildlife monitoring to infrastructure or asset management. Within the robot localization itself, there is a taxonomy of problems to be solved based on how one looks at the problem [1]. One such category is based on type of the initial knowledge that a robot may possess initially and/or at run-time. There are three problems in this category; position tracking, global localization [2,3], and kidnapped problem [4,5]. Position tracking is a problem to localize a mobile robot based on the information from the environment and previous knowledge of the pose. Global localization is similar to pose tracking, however there is no previous pose knowledge available as discussed in [6] and [7]. Kidnapped Robot Problem (KRP) is more difficult than the two. This problem is defined as a condition when the robot is being transported somewhere in the map while doing localization or pose tracking, without knowing where the destination is and without even realizing it has been moved.

KRP rarely happens in real life, however the ability to solve KRP is usually considered as a measure of good localization technique. Another importance of KRP study is the concern on robot's safety. An undetected and/or unsolved KRP leads to incorrect map and incorrect pose estimate, thus may hinder the robot from performing its task. A more dangerous problem is when the robot wanders around an undefined/incomplete map while believing that it is still performing localization well. This condition is dangerous when there is a potential hazard in the undefined/incomplete map. The kidnapping event may happen at any time during indoor robot's exploration [8]. Therefore, an accurate online kidnapping detector is preferable in this case, such that there is immediate and reliable information that the robot is kidnapped at any points in time. This type of detector will allow precautionary action to be taken right after kidnapping, and determine whether a global localization process is required.

This paper proposes an augmented online approach for a mobile robot to effectively detect the kidnapping event in range-finder-based indoor localization, based on shape similarity of environment scans between two consecutive time instances. The shape similarity measure used in the approach is a variant of Two-Dimensional Dynamic Time Warping. The rest of the paper is organized as follows. In Section 2, a literature review on existing detection methods is presented. Section 3 lays the groundwork and details of the proposed detection method while the details of benchmarks are presented in Section 4. Section 5 discusses the results of proposed online detection approach and lastly Section 6 concludes the paper.

## 2. Related Works for Kidnapping Problem in Localization

Thrun et al. in their work of Augmented Monte Carlo Localization [3] solves KRP by injecting random new hypotheses (called particles) of a robot's pose at every time step. By this procedure, the robot can relocalize after kidnapping without detecting the event. This offline approach however is not preferable since one needs to wait until the end of a robot's exploration to obtain kidnapping information. The other problem with this approach is that it is inefficient because the relocalization process is executed at every time step even when it is unnecessary, i.e., no kidnapping event occurred at that particular time step.

In order to achieve better KRP solution, numerous online detection methods of kidnapping events have been proposed. Zhang et al. in [9] detect kidnapping by using the maximum weight of current set particles (pose hypotheses) in Monte Carlo Localization. A similar parameter is proposed by Choi et al. in [5], which uses the entropy of the particles' weight instead of the maximum weight. These two approaches solved the online inabilities of Augmented Monte Carlo Localization; however, the accuracy of detection is still not of concern.

Metric-based detection by Campbell et al. [4] address the accuracy of detection more specifically. The approach is based on a new technique of deterministic localization based on scan matching introduced in [10], called Normal Distribution Transform (NDT). Their four detection metrics are based on the error between two scans aligned by the localization (NDT in their work), likelihood of a scan lies on the NDT plane of the reference scan, uncertainty of the least known pose degree of freedom, and the error between NDT-transformed and odometry transformed point clouds. It has not been proven, however, that the method maintains high accuracy to detect kidnapping events across all time instances of a mobile robot's exploration time. This paper proposed a new approach to detect kidnapping events accurately across all time instances of a robot's exploration. The underlying idea is that one can determine when it is being kidnapped by measuring the similarity of the environment seen between two consecutive time instances. Given a small natural movement at each time instance, a significant change in environment can be an indication that it is being moved unnaturally, such as slipping or being taken and 'woken up' somewhere else. Based on this idea, the problem is thus reduced to shape similarity between the environment scans at two consecutive time instances. A high similarity score

indicates the natural movement, and vice versa. An augmented variant of 2-D Dynamic Time Warping is used as a similarity measure.

Indoor mobile robotics with an onboard computational capacity and a better payload can easily incorporate a light two-dimensional (2D) laser rangefinder, which will be considered as an additional sensor that will greatly improve the results of the localization and mapping system under certain environment conditions. Previous studies on 2D indoor localization focus towards the laser SLAM and visual SLAM fusion to provide robust localization [11]. Hess et al. also report the use of portable laser range finders for generating and visualizing floor plans in real-time [12]. Recently, a new sensor fusion method for visual SLAM was proposed through integration of a monocular camera and a 1D-laser range finder where it overcomes the limited depth range problem associated with SLAM for RGBD cameras [13]. Several approaches of online kidnapping detection have been proposed over the years [4–6]. In this paper, we investigate their detection performance across all kidnapping points in feature-less corridor map within a Gazebo environment such as the problem addressed in [4] and [9]. It can be defined as a good detector with these criteria:

(1) The detector should be an online detector [4], such that it may reduce the possibility that the drone wanders around into a possible hazardous point in the map after kidnapping, because an immediate preventive action might be taken if the kidnapping information is obtained immediately after the kidnapping event.

(2) The detector should be able to differentiate normal condition from kidnapping condition as well as possible, i.e., there is a wide gap in the detector's output when the robot is in normal condition compared to when it is being kidnapped. This criterion should apply to a wide range of kidnapping points (time instances at which kidnapping events might occur). The wide time range application of a detector is important because a kidnapping may occur anytime during a robot's exploration.

## 3. Quasi-Standardized 2-D Dynamic Time Warping

In this section, our proposed approach to detect kidnapping event in feature-less map is presented. The underlying idea is that one can realize a kidnapping event happening when there is a significant change in the perceived environment. In feature-less map, the only information a robot can obtain is the distance to the walls within the range of the sensor. A point cloud can be constructed from the distance measured by each sensor beam [14]. The problem of kidnapping detection can thus be simplified to measuring the similarity between two geometric shapes formed by the point clouds. This paper proposes a novel variant of Multidimensional Dynamic Time Warping (MDDTW) for a two-dimensional case called Quasi-Standardized 2-D Dynamic Time Warping (QS-2DDTW). Before presenting the details of the proposed method, an understanding on a few key aspects of the method are necessary, namely the standard Dynamic Time Warping, Multidimensional Dynamic Time Warping, and the measure of variance similarity.

### 3.1. Dynamic Time Warping

Dynamic Time Warping (DTW) is a popular temporal sequence alignment method [15], with applications ranging in speech recognition [16,17], human motion animation [18], human activity recognition [19], and time series classification [20]. One key ability of DTW which distinguished it from other alignment method lies in its ability to nonlinearly align two sequences, such that two similar signals will have small distance even though one of the signals is distorted or 'warped' in time, for example, by shifting it a few steps ahead [21]. A comparison of alignment result between Euclidean distance, which measures distance between $i$-th point in one signal to the $i$-th point in another signal, and DTW can be seen in Figure 1.

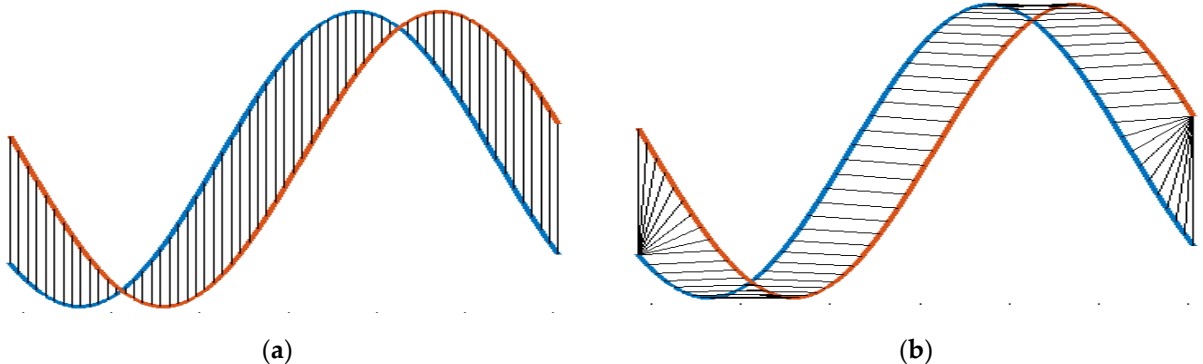

**Figure 1.** Alignment between two signals differs in phase. (**a**) is the result of Euclidean distance alignment, while (**b**) shows the resulting DTW.

It can be clearly seen from the figure that alignment using DTW provides a better measure of similarity since it aligns similar features together (peak vs. peak, trough vs. trough). Similarity measures using DTW is suitable for shape similarity approach in our detection method. This is because the reading of each sensor is not maintained between two consecutive time steps, however, the general shape of the point cloud, i.e., the 'shape' of the environment is more or less maintained provided that the robot moves along a sufficiently small distance between any two consecutive time steps [14].

Given $\mathcal{P}$ and $\mathcal{Q}$, two sequences of real-valued univariate time series of length $\mathcal{L}_\mathcal{P}$ and $\mathcal{L}_\mathcal{Q}$, that is $\mathcal{P} = (p_1,\ p_2,\ \ldots,\ p_{\mathcal{L}_\mathcal{P}})$ and $\mathcal{Q} = \left(q_1,\ q_2,\ \ldots,\ q_{\mathcal{L}_\mathcal{Q}}\right)$. Let $\mathcal{D}(\mathcal{P},\mathcal{Q}) \in \mathcal{R}^{\mathcal{L}_\mathcal{P} \times \mathcal{L}_\mathcal{Q}}$ represent the pairwise distance matrix between $\mathcal{P}$ and $\mathcal{Q}$ such that $\mathcal{D}(\mathcal{P},\mathcal{Q})_{i,j} = \left|p_i - q_j\right|$. The alignment problem is to find the sequences of indices $\mathcal{I}_\mathcal{P}$ and $\mathcal{I}_\mathcal{Q}$ of the same length $\ell(\ell \geq \max(\mathcal{L}_\mathcal{P}, \mathcal{L}_\mathcal{Q}))$ which match index $\mathcal{I}_\mathcal{P}(i)$ in $\mathcal{P}$ and $\mathcal{I}_\mathcal{Q}(i)$ in $\mathcal{Q}$ that minimizes the cost $\mathcal{C} = \sum_{i=1}^{l} \mathcal{D}(\mathcal{P},\mathcal{Q})_{\mathcal{I}_\mathcal{P}(i),\mathcal{I}_\mathcal{Q}(i)}$.

REMARK 1: The choice of path to minimize $\mathcal{C}$ should fulfill three constraints. The first one is *Monotonicity* constraint, which states that the path will never return on itself. Both matching indices will either stay the same or increase; they never decrease. The second one is *Continuity* constraint, which means the path should only advance one step at a time, that is, both indices increase at most by one every step. The last constraint is *Boundary constraint* which restricts the path to start from bottom left corner (initial time index of the sequence) to the top right corner (final time index of the sequence) [22].

### 3.2. Multidimensional Dynamic Time Warping

Holt et al. introduced a variant of Dynamic Time Warping to work with multidimensional time series in [19]. Let $\mathcal{P}$ and $\mathcal{Q}$ be two real-value sequences of time series with dimension $K$ and length $\mathcal{L}_\mathcal{P}$ and $\mathcal{L}_\mathcal{Q}$, respectively. The algorithm of MD-DTW is summarized in Algorithm 1.

---

**Algorithm 1** MD-DTW algorithm

---

1. **MD-DTW Algorithm**
2. Normalize each dimension of $\mathcal{P}$ and $\mathcal{Q}$ separately to zero mean and unit variance
3. Construct distance matrix $\mathcal{D}(\mathcal{P},\mathcal{Q}) \in \mathcal{R}^{\mathcal{L}_\mathcal{P} \times \mathcal{L}_\mathcal{Q}}$ according to
4. $\mathcal{D}(\mathcal{P},\mathcal{Q})_{i,j} = \sum_{k=1}^{K} \left|p_{i,k} - q_{j,k}\right|$
5. Find the minimum cost path using the regular DTW under the constraints stated in Remark 1 on this distance matrix

---

### 3.3. Single and Two-Dimensional Dynamic Time Warping

The problem of detection at hand can technically be approached using 1-D DTW or 2-D DTW depending on how environment reading is represented. Let $D^t = \left\{d_1^t, d_2^t, \ldots, d_S^t\right\}$ represent the set of distance reading of $S$ sensor beams at time $t$. For 1-D approach, direct

use of $D$ as the signals to aligned can be applied, i.e., $\mathcal{P} = D^{t-1}$ and $\mathcal{Q} = D^t$. For 2-D approach, the signals consist of 2-D point clouds $\mathcal{P} = \begin{bmatrix} \mathcal{P}_x & \mathcal{P}_y \end{bmatrix}$ and $\mathcal{Q} = \begin{bmatrix} \mathcal{Q}_x & \mathcal{Q}_y \end{bmatrix}$ extracted from $D^{t-1}$ and $D^t$, respectively, by the following function

$$\mathcal{P} = \begin{bmatrix} D^{t-1}\cos(\varphi) & D^{t-1}\sin(\varphi) \end{bmatrix}, \; \mathcal{Q} = \begin{bmatrix} D^t\cos(\varphi) & D^t\sin(\varphi) \end{bmatrix} \tag{1}$$

where $\varphi$ is the angle of each sensor beam $s$ which corresponds to $d_s^t$ with respect to a robot's local coordinate frame. In this section, we investigate the performance of the two approaches in solving kidnapping detection. Let $\mathcal{M}$ define the detection parameter value of an arbitrary kidnapping detector method ($\mathcal{M}$ can be particles weight, similarity degree, measurement entropy, etc., depending on the detector). For each detector, the detector performance evaluator of each trial is expressed as

$$\mathcal{F}(\mathcal{M}) = \begin{cases} 1 - \left( \max_{2 \leq t \leq tk-1}(\mathcal{M})/\mathcal{M}_{tk} \right), & \mathcal{M}_t \text{ is HPTF} \\ 1 - \left( \mathcal{M}_{tk}/\min_{2 \leq t \leq tk-1}(\mathcal{M}) \right), & \mathcal{M}_t \text{ is LPTF} \end{cases} \tag{2}$$

where $t$ is the time interval of the robot's operation and $2 \leq tk \leq T$ is the kidnapping points, i.e., the time instance at which kidnapping event occurred with kidnapping destination. LPTF is a short form of Low Pass Threshold Function, which we define as a threshold function in which the function holds true when the value is lower than the specified threshold. Similarly, we define HPTF (High Pass Threshold Function) as a threshold function in which the function holds true when the value is higher than the specified threshold. This performance evaluator is intended to measure the ability of $\mathcal{M}$ to distinguish between the kidnapping condition and the normal condition around the kidnapping points, based on how far the value at $t = t\_k$ from the value at any other prior points. Negative value of $\mathcal{F}(\mathcal{M})$ implies that there is another value(s) of $\mathcal{M}_{\{2 \leq t \leq t\_k-1\}}$ higher (for HPTF) or lower (for LPTF) than $\mathcal{M}_{t\_k}$ itself, indicating a misdetection for kidnapping at $t = t\_k$.

In order to obtain the performance index of 1D-DTW and 2D-DTW, a 20 trials performance test is executed and the complete algorithm for the performance test is shown in Algorithm 2 below.

---

**Algorithm 2** Performance test algorithm

---

1. $\mathcal{M}$ **Performance Test Algorithm**
2. For all $2 \leq t\_k \leq 100$, do
3.     Let the robot run normally. At $t = t\_k$ the robot is kidnapped to a certain place and after that it continues to run normally until $t = 100$
4.     See if the robot can detect that it has been kidnapped at $t = t$ by measuring $\mathcal{F}(\mathcal{M})$
5. Add $\mathcal{F}(\mathcal{M})$ to a vector of size $[20 \times 1]$ $\eta$
6. Repeat this 20 times and measure the performance index $\overline{\eta}$ by averaging $\eta$

---

Therefore, mean of performance index is defined as

$$\overline{\eta} = \frac{1}{N}\sum_{n=1}^{N} \eta_{n, 2 \leq t\_k \leq T} \tag{3}$$

Summary of the test result is depicted in Figure 2 and it is obviously apparent that raw (unstandardized) 2-D DTW performs better than unstandardized 1-D DTW. The drop at around $t = 30$ is caused by similarity in sensor reading between the location of the robot before and after kidnapping destination (around $t = 30$ the robot is at a point in the map which gives similar rangefinder reading as the kidnapping destination reading). As can be seen in Figure 2, the 2-D versions perform better at distinguishing in this very similar sensor reading. Moreover, a few misdetections happen towards the end even for 2-D DTW. This can be mitigated by introducing the standardization as in the algorithm in Algorithm 1; however, it generally reduces the performance compared to the unstandardized version.

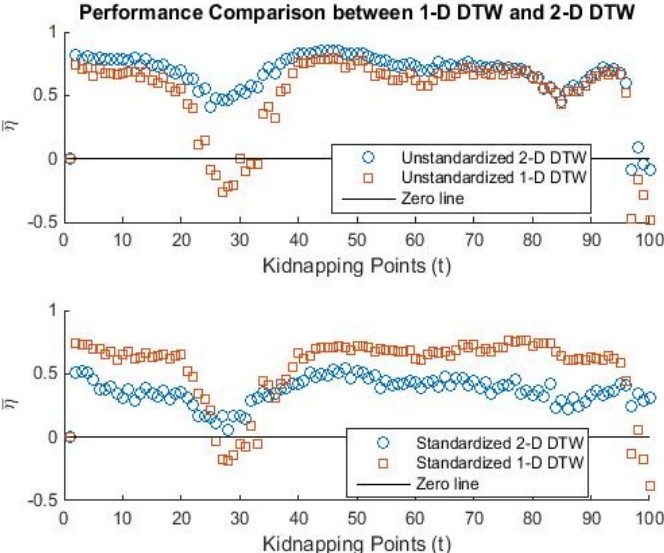

**Figure 2.** Comparison of performance between 1-D DTW and 2-D DTW. The top plot shows the comparison between the unstandardized versions of the two, while the bottom one shows the comparison between the standardized versions of them.

Another tool is thus required to ensure high performance while reducing the number of misdetections.

### 3.4. Variance Homogeneity Test

Variance homogeneity test is a test to verify whether the *k* samples have equal variances [23,24]. In other words, it is used to test the null hypothesis that the *k* samples are drawn from the same distribution. Based on the assumption of the distribution the samples came from, the test can be divided into two types; normal distribution-based test and non-normal distribution-based test:

Bartlett Test is one popular test of variance homogeneity [25]. It has high sensitivity towards the non-normality, thus, it works best if the distribution the sample came from is the normal distribution. The test statistic is defined as

$$V = \frac{(N-k)\ln s_p^2 - \sum_{i=1}^{k}(N_i - 1)\ln s_i^2}{1 + \left(\frac{1}{(3(k-1))}\right)\left(\left(\sum_{i=1}^{k}\frac{1}{(N_i-1)}\right) - \frac{1}{(N-k)}\right)} \tag{4}$$

where $s_i^2$ is the variance of the *i*th group, $N$ is the sample size of the *i*th group, $k$ is the number of groups, and $s_p^2$ is the pooled variance, defined as

$$s_p^2 = \sum_{i=1}^{k} \frac{(N_i - 1)s_i^2}{(N-k)} \tag{5}$$

Levene Test [26] is a homogeneity test with lower sensitivity towards non-normality, thus it works better than the Bartlett Test if the samples are drawn from non-normal distribution. The test statistic, $V$ is defined as follows

$$V = \frac{(N-k)\sum_{i=1}^{k} N_i \left(\overline{Z}_{i.} - \overline{Z}_{..}\right)^2}{(k-1)\sum_{i=1}^{k}\sum_{j=1}^{N_i}\left(Z_{ij} - \overline{Z}_{i.}\right)^2} \tag{6}$$

where $\overline{Z}_{i.}$ is the group means of $Z_{ij}$ and $\overline{Z}_{..}$ is the overall mean of $Z_{ij}$. $Z_{ij}$ can be represented in a variety of different definitions, however, the one that is known as more robust than the other is called the Brown–Forsythe test which defines $Z_{ij}$ in terms of the median, that is

$$Z_{ij} = \left| Y_{ij} - \overline{Y}_{i.} \right| \tag{7}$$

where $\overline{Y}_{i.}$ is the median of $i$th subgroup.

Variance homogeneity tests accept the null hypothesis that all $k$ samples are drawn from the distributions with the same variance if

$$V < F_{\alpha,k-1,N-k} \tag{8}$$

where $F_{\alpha,k-1,N-k}$ is the upper critical value of $F$-distribution with $k-1$ and $N-k$ degrees of freedom at significance level $\alpha$.

### 3.5. Quasi-Standardized Two-Dimensional Dynamic Time Warping

In this paper, we introduce a new variant of Dynamic Time Warping for two-dimensional time series called Quasi-Standardized Two-Dimensional Dynamic Time Warping (QS-2DDTW) as similarity measure of environment scan between two consecutive time steps, which acts as a kidnapping detector. Let $\mathcal{P}$ and $\mathcal{Q}$ define two point clouds extracted based on the reading of environment by range finder sensor at two time instances $t-1$ and $t$, respectively. The complete algorithm QS-2DDTW is shown in Algorithm 3.

---

**Algorithm 3** QS-2DDTW algorithm

---

1. **QS-2DDTW Algorithm**
2. Decompose each dimension of $\mathcal{P} = \begin{bmatrix} \mathcal{P}_x & \mathcal{P}_y \end{bmatrix}$ and $\mathcal{Q} = \begin{bmatrix} \mathcal{Q}_x & \mathcal{Q}_y \end{bmatrix}$
3. Perform variance homogeneity test (dedicated test to be employed depends on normality of the noise) on both dimensions

$$V_x = V(\mathcal{P}_x, \mathcal{Q}_x)$$
$$V_y = V(\mathcal{P}_y, \mathcal{Q}_y)$$

4. **if $V_x < \alpha$ and $V_y < \alpha$ and $V_x > 1$ and $V_y > 1$**
5. $\mathcal{P} = \begin{bmatrix} \dfrac{\mathcal{P}_x - \overline{\mathcal{P}}_x}{\sigma_{\mathcal{P}_x}} & \dfrac{\mathcal{P}_y - \overline{\mathcal{P}}_y}{\sigma_{\mathcal{P}_y}} \end{bmatrix}$
6. $\mathcal{Q} = \begin{bmatrix} \dfrac{\mathcal{Q}_x - \overline{\mathcal{Q}}_x}{\sigma_{\mathcal{Q}_x}} & \dfrac{\mathcal{Q}_y - \overline{\mathcal{Q}}_y}{\sigma_{\mathcal{Q}_y}} \end{bmatrix}$
7. **End if**
8. Construct distance matrix $\mathcal{D}(\mathcal{P}, \mathcal{Q}) \in \mathcal{R}^{\mathcal{L}_{\mathcal{P}} \times \mathcal{L}_{\mathcal{Q}}}$ according to

$$\mathcal{D}(\mathcal{P}, \mathcal{Q})_{i,j} = \sum_{k=1}^{2} \left| p_{i,k} - q_{j,k} \right|$$

9. Find the minimum cost path using the regular DTW under the constraints stated in Remark 1 on this distance matrix.

---

The introduction of variance homogeneity test in line 3 has a purpose to selectively decide when to do standardization and when not to do it. This selection is an attempt to improve DTW detector's ability to distinguish between normal and kidnapping condition, i.e., increasing their performance index. Please note that the choice of $V$ depends on the estimated normality of measurement noise. In order to verify the improvement, the same performance test as in Algorithm 2 is performed. The result of the test is shown in Figure 3.

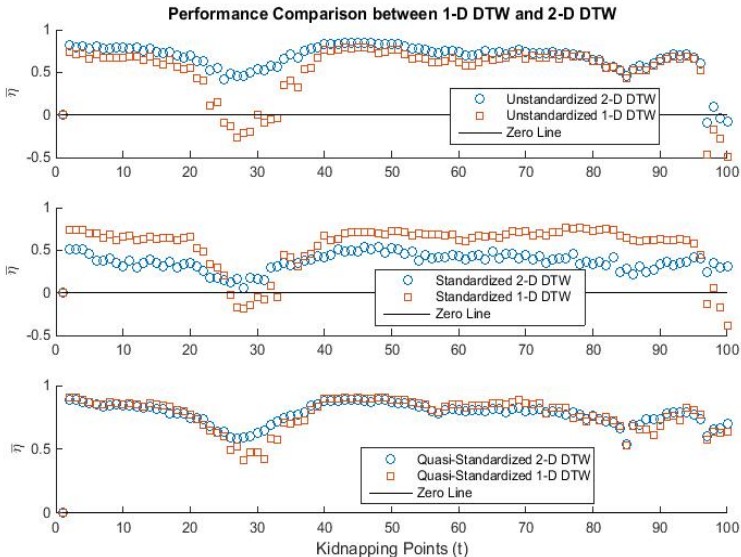

**Figure 3.** Comparison of performance between unstandardized 1-D DTW and unstandardized 2-D DTW (**Top**), standardized 1D-DTW and standardized 2-D DTW (**Middle**), and QS1DTW and QS2DTW (**Bottom**).

The result of unstandardized and standardized versions is also shown for comparison purposes. It is clear from the figure that introducing selection of standardization improves the performance of both 1-D DTW and 2-D DTW as indicated by the absence of negative values while maintaining the high performance level of the unstandardized version. Among these two approaches, however, QS-2DDTW still outperforms QS-1DDTW.

## 4. Comparative Analysis with the Existing Online Detection Approaches

In this section, basic understanding of three existing online detectors is briefly presented.

### 4.1. Maximum Current Weight

The detector is proposed in [9] by Zhang et al. The detector works under the Monte Carlo Localization (MCL) [27], which uses samples of possible poses to approximate a robot's pose belief. These samples (called particles in MCL terminology) are weighted based on how close the environment reading given by each sample is to the reading of the robot, i.e., the higher the weight the better the sample in representing the robot's true pose. At each time step, the maximum weight of the current set of particles is calculated. A kidnapping event is determined by comparing this value against a threshold. Mathematically, it can be written as

$$Kidnapped_t = \begin{cases} 1 & \omega_t^{max} < \gamma \\ 0 & Otherwise \end{cases} \tag{9}$$

### 4.2. Measurement Entropy

Choi et al. in [5] also define a kidnapping detector under Monte Carlo Localization in a topological map with recognizable objects/landmarks at each node. In their work, a metric called measurement entropy, is defined as follows

$$H_t(p) = -\sum_{x_t^{[y]}} p\left(s_t, o_t, z_t \middle| x_t^{[y]}, m\right) \log p\left(s_t, o_t, z_t \middle| x_t^{[y]}, m\right) \tag{10}$$

where $s_t$, $o_t$, $z_t$ are the distance context to objects at time $t$, objects seen by the robot at time $t$, and the features extracted from the objects at time $t$, respectively. $x_t^{[y]}$ is the state of particle $y$ at time $t$ and $m$ defines the map.

It can be seen that the term inside the summand in Equation (2) is in fact the weight of particle $y$ and thus the equation can be described as the sum of a particle's weight entropy, that is

$$H_t(p) = -\sum_{x_t^{[y]}} \omega_t^{[y]} log \omega_t^{[y]} \tag{11}$$

Kidnapping detection is based on the rise of this measurement entropy, which can be written as

$$Kidnapped_t = \begin{cases} 1 & H_t(p) \geq \pi \\ 0 & Otherwise \end{cases} \tag{12}$$

where $\pi$ is some constant as a threshold to distinguish kidnapping from normal condition.

*4.3. Metric-Based Detector*

Recently, Campbell et al. in [4] defined a set of four metrics to detect a kidnapping event. Their work is mainly derived under a deterministic scan matching-based localization called Normal Distribution Transform (NDT) introduced by Biber and Strasser in [10]. Then, Magnusson et al. extended NDT in three-dimensional space [28] and [29].

Given two sequential point clouds extracted from environment scan $\mathcal{P}$ and $\mathcal{Q}$, matrix ${}_{\mathcal{Q}}^{\mathcal{P}}T$ estimated by NDT, which transforms coordinate frame $\mathcal{Q}$ into coordinate frame of $\mathcal{P}$, and the same transformation estimated by the odometry ${}_{\mathcal{Q}}^{\mathcal{P}}U$, the four metrics are defined as follow:

i.   Mean squared error between the two NDT-aligned point clouds, which is written as

$$Q_e\left(\mathcal{P}, \mathcal{Q}, {}_{\mathcal{Q}}^{\mathcal{P}}T\right) = \frac{1}{n} \sum_{i=1}^n \|\vec{p}_i - \vec{q}_i\|^2 \tag{13}$$

where $n$ is the number of overlapping points, $\vec{p}_i$ is a point in $\mathcal{P}$, and $\vec{q}_i$ the nearest-neighbor point of $\vec{p}_i$.

ii.  The likelihood that the transformed point cloud $\mathcal{Q}$ lies on the surface of NDT of $\mathcal{P}$, that is

$$Q_s\left(\mathcal{P}, \mathcal{Q}, {}_{\mathcal{Q}}^{\mathcal{P}}T\right) = \frac{1}{n} \sum_{i=1}^n \check{p}\left({}_{\mathcal{Q}}^{\mathcal{P}}T, \vec{b}_i\right) \tag{14}$$

where $n$ is the number of points of $\vec{b}_i$ in $\mathcal{Q}$.

iii. Standard deviation from the least known pose degree of freedom, calculated from maximum eigenvalue $\lambda$ of NDT optimization function.

$$Q_h\left(\mathcal{P}, \mathcal{Q}, {}_{\mathcal{Q}}^{\mathcal{P}}T\right) = \sqrt{\max_{i=1:v} \lambda_i} \tag{15}$$

where $v$ is degree of freedom of a robot's pose.

iv.  Mean squared error of point clouds $\mathcal{Q}$ transformed by ${}_{\mathcal{Q}}^{\mathcal{P}}T$ and ${}_{\mathcal{Q}}^{\mathcal{P}}U$,

$$Q_h\left(\mathcal{Q}, {}_{\mathcal{Q}}^{\mathcal{P}}T, {}_{\mathcal{Q}}^{\mathcal{P}}U\right) = \frac{1}{n} \sum_{i=1}^n \|\vec{q}_i - \vec{r}_i\|^2 \tag{16}$$

**5. Results and Discussion**

In this section, a series of simulations that include laser measurements from the Turtlebot3 platform is performed to validate our proposed online detection approach. A unit of TurtleBot3 (see in Figure 4) consisting of a two-wheeled platform where its weight and size are 1.8 kg and $281 \times 306 \times 141$ mm, respectively, is selected.

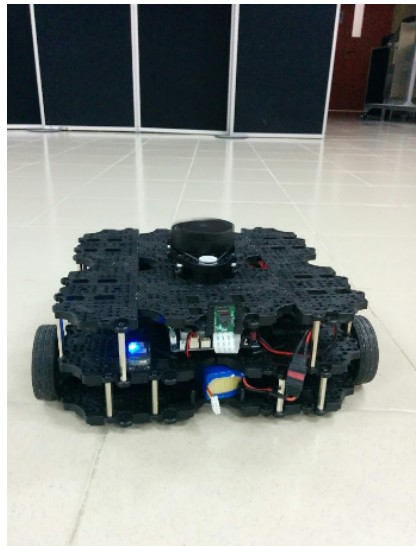 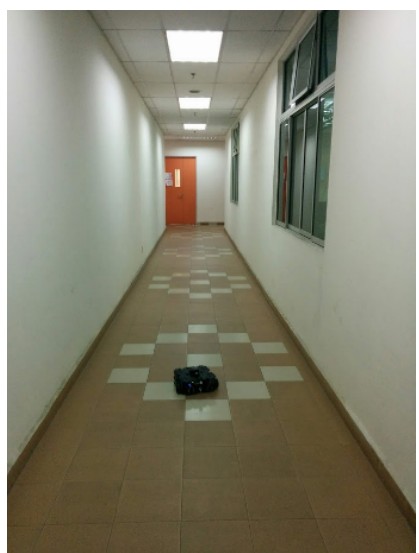

**Figure 4.** TurtleBot3 Waffle platform (**left**) and its navigation through an indoor area (**right**).

The mobile robot runs with a use single-board computer with the latest version of Ubuntu Linux (16.04.2 LTS) and Robot Operating System (ROS) Kinetic. The development of ROS packages is still ongoing where in [30], researchers proposed a ROS wrapper of Real Time Appearance Based Mapping with the help of a low cost camera and the Microsoft Kinect XBOX 360 mounted on Turtlebot to improvise the indoor navigation. Moreover, Adaptive Monte Carlo Localization was used for localizing the Turtlebot model in the map. Recent work by Ladosz et al. proposed a ROS framework for development and testing of autonomous control functions. The flexibility and power of ROS was combined with the Robotic Systems toolbox from MATLAB/Simulink, Linux embedded systems and a commercially available autopilot. Thus, it results in a low cost, simple, highly flexible, and reconfigurable system [31].

A 360°LiDAR_node ROS package can be added easily to read and transmit the laser measurements to the remote processor. The use of LiDAR sensor Turtlebot3 can be accessed and commanded from ROS using the mavros package. This TurtleBot model also has a Raspberry Pi Camera and multiple sensors, including gyroscopic, magnetometer, and accelerometer. Note that the system model platform which is running on Gazebo environment is very close to the actual experimental [32], and the multimodal components in the simulated environment can reflect the characteristics of the actual components in a more comprehensive way.

It was compared with three benchmarks, namely Maximum Current Weight (MCW), Measurement Entropy (ME), and Metric-based Detector against QS-2DDTW with the same test as in Algorithm 2. Two types of test were performed which were "Test with re-localization" and "Test without re-localization". The purpose of this test division was to see how dependent the accuracy of $\mathcal{M}$ was to the relocalization process which was started at $t = t\_k + 1$. For this purpose, the evaluator in (2) was evaluated for all time instances $2 \leq t \leq T$ instead of $2 \leq t \leq t\_k - 1$. White Gaussian noise with variance 0.1 was employed in the measurement model in each simulation. Since the Gaussian noise model was used, Bartlett test for the variance test was employed for the proposed method. The kidnapping scenario is depicted in Figure 5. This kidnapping scenario can be explained more explicitly, compactified from Algorithm 2, as below:

For all $2 \leq t\_k \leq 100$, perform the following procedure:

i.    Let the robot run normally. At $t = t\_k$ the robot is kidnapped to a place indicated in Figure 5 and after that it continues to run normally until $t = 100$ (at $t = t\_k + 1$ the robot can be relocalized or not, depending on the type of test)

ii.    See if the robot can detect that it has been kidnapped at $t = t\_k$.

    iii.    Repeat this 100 times and measure the performance index.

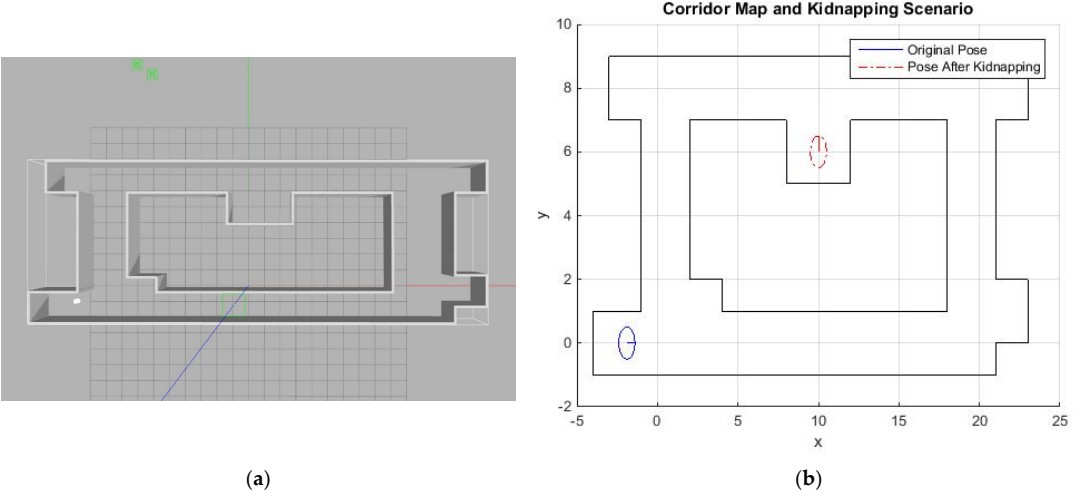

| (**a**) | (**b**) |

**Figure 5.** A corridor map used in the Gazebo (**a**) and the instance of a kidnapping event (**b**).

### 5.1. Performance Evaluation against MCW and ME

For these MCL-based detectors, the simulation was run using 2000 particles. No relocalization process was implemented and the localization was performed in feature-less corridor map as shown in Figure 5. The performance of the two detectors against our proposed method without and with relocalization is depicted in Figures 6 and 7, respectively.

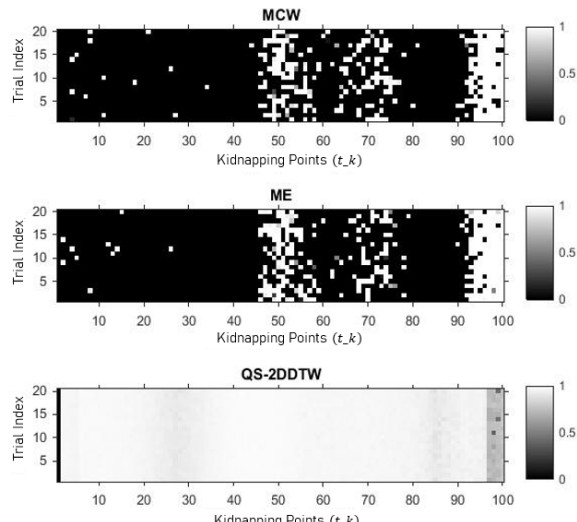

**Figure 6.** Performance results of MCW (**top**), ME (**middle**), and proposed QS-2DDTW (**bottom**) without relocalization. Black pixels indicate $\eta \leq 0$.

It is shown clearly on Figure 6 where the black pixels indicate $\eta \leq 0$ that under no relocalization policy, MCW and ME perform very poorly. The huge improvement of these two methods when the test is changed to employ relocalization (as shown in Figure 7) indicates that the two methods depend very heavily on the success of the relocalization process. The relatively high dependency is caused by the fact that when relocalization fails (or for the extreme case, none is implemented) the weight of particles exhibit similar low values as in the kidnapping case, inducing detections when no kidnapping event occurs. On the other hand, QS-2DDTW is independent towards the localization process, thus it can maintain a high performance in both tests (with and without relocalization).

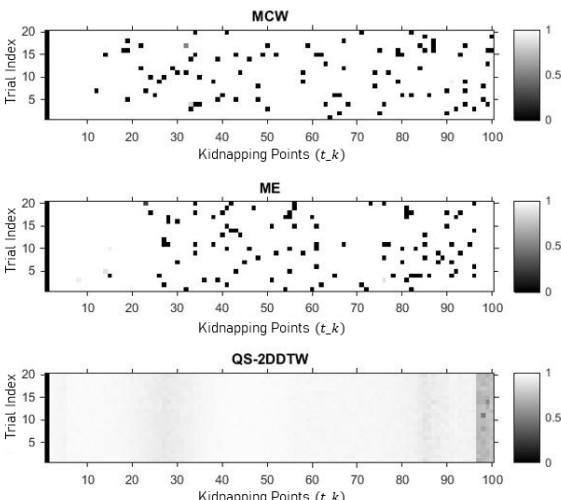

**Figure 7.** Performance result of MCW (**top**), ME (**middle**), and proposed QS-2DDTW (**bottom**) with relocalization. Black pixels indicate $\eta \leq 0$.

### 5.2. Performance Evaluation against Metric-Based Detector

In this section, there are four metrics to be tested, namely **Qe**, **Qs**, **Qh**, and **Qo**. Similar to our proposed approach, these metrics are *local*, i.e., it depends only on two consecutive time instances by comparing the scans obtained by the sensor as given in (5) to (8). The performance of these metrics against QS-2DDTW without and with relocalization are shown in Figures 8 and 9, respectively.

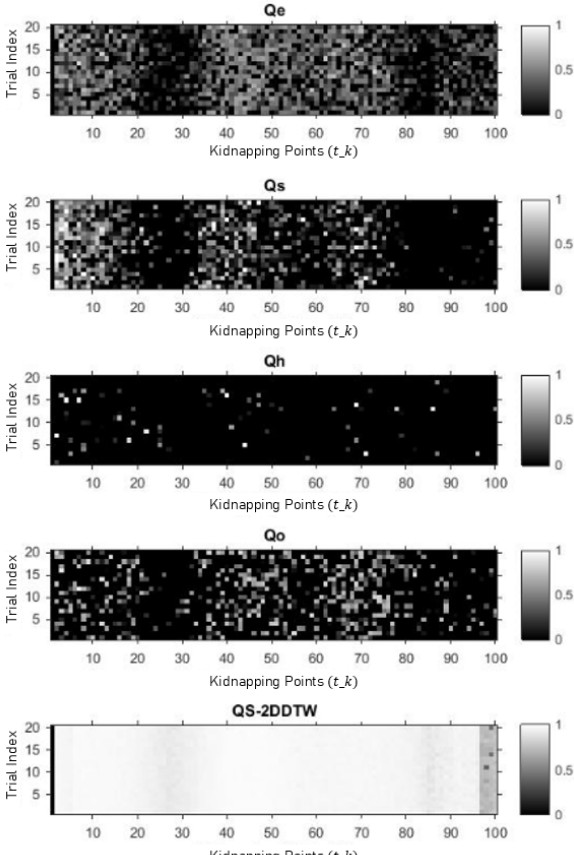

**Figure 8.** Performance result of metric-based detectors against QS-2DDTW (**bottom**) without relocalization. Black pixels indicate $\eta \leq 0$.

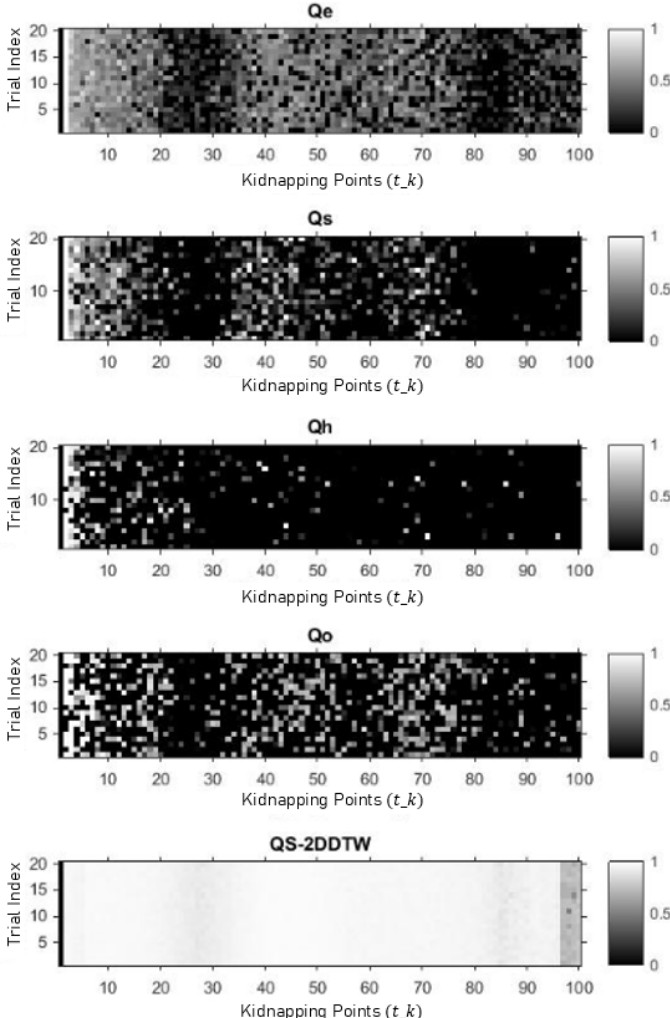

**Figure 9.** Performance result of metric-based detectors against QS-2DDTW (**bottom**) with relocalization. Black pixels indicate $\eta \leq 0$.

From the presented results in Figures 8 and 9, the small changes in metrics-based detectors between the first test (without relocalization) and second test (with relocalization) indicate a low dependency towards relocalization, thus the detection accuracy is very less likely affected by the success rate in the relocalization process.

Among the five detectors, however, QS-2DDTW once again outperforms others in terms of accuracy for kidnapping across all time instances. It can successfully distinguish normal conditions around kidnapping points from the kidnapping event at the kidnapping points itself better than other methods used as benchmarks. A low dependency towards relocalization also ensures that the accuracy is maintained no matter how successfully the relocalization can be achieved.

Different experiments were set up to evaluate the effectiveness as well as demonstrate the performance of proposed QS-2DDTW in real static and featureless indoor environments as shown in Figure 10. The map of the environment treated as a ground truth with several room dividers and walkway exists in this environment. In Figure 10a, the robot is performing a 2D area mapping while Figure 10b shows the map generated in ROS visualization (RViz) from those indoor environments. An experimental analysis was then performed in several kidnapping and nonkidnapping situations under Metric-Based Detector and the proposed QS-2DDTW. The classical statistical analysis was carried out to examine the difference in the distribution of variables between the presence and absence of kidnapping. Nominal variables (kidnapping time, distance between the start and the end points of the kidnapping event and robot orientation) were independently important

between presence and absence of kidnapping. Using the independent predictors, we used LR technique with a stepwise variable selection and receiver operating characteristics (ROC) curve to visualize the probability of presence of kidnapping. Note that the ROC curve is based on the true positive rate that represents a fraction of the detected kidnapping out of the total number of actual kidnapping events as well as the false positive rate that is the fraction of the incorrectly detected nonkidnapping out of the total number of actual nonkidnapping events.

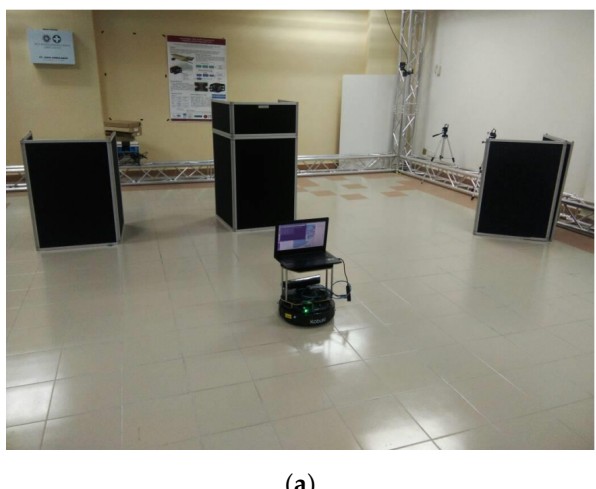
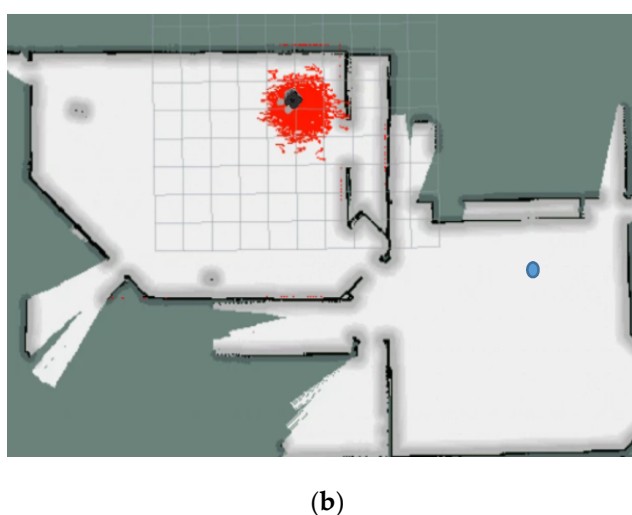

(**a**)          (**b**)

**Figure 10.** The mapping process and result of experimental testbed. (**a**) Mapping process using Turtlebot, (**b**) mapping result.

In the initial experiment, we evaluated the detection method under a nonkidnapping situation where there is some possibility of false alarm. A case study of a nonkidnapping situation with Gmapping in this environment as shown in Figure 11 also helps to ensure the smoothness of robot navigation (collision free) as well as validate the recorded coordinates (its starting and ending points).

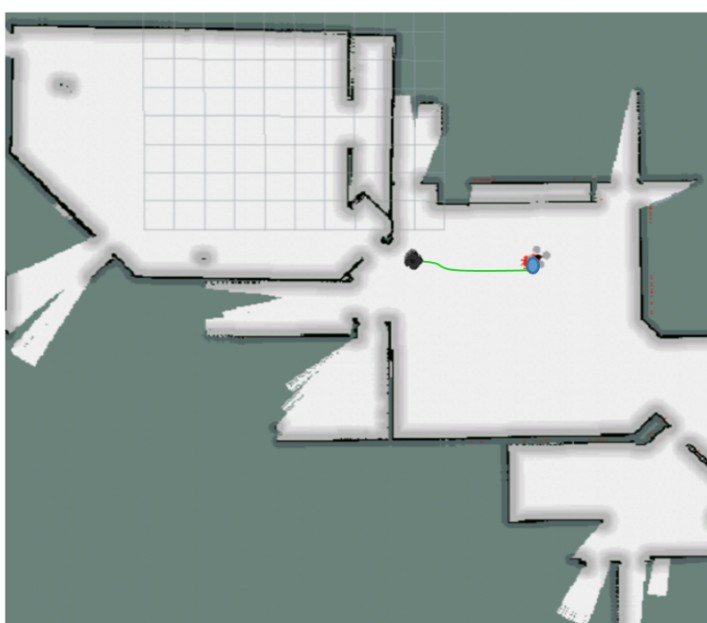

**Figure 11.** Experimental case study of mapping result of a nonkidnapping situation.

The second experimental case study involves a robot kidnapping situation as shown in Figure 12. From the mapping result created by Gmapping, Figure 12 depicts the starting

and ending locations of the kidnapping event. First the mobile robot runs normally where it moves from the start point (blue point) along the green trajectory until $t = t\_k$ at which kidnapping begins. At random time $t = t\_k$ the mobile robot is kidnapped, and it is observed whether the proposed QS-2DDTW is able to detect the robot kidnapping at $t = t\_k$. This process will be repeated few times under variation of independent predictors before analyzing the performance of the detection algorithm based on LR technique and ROC.

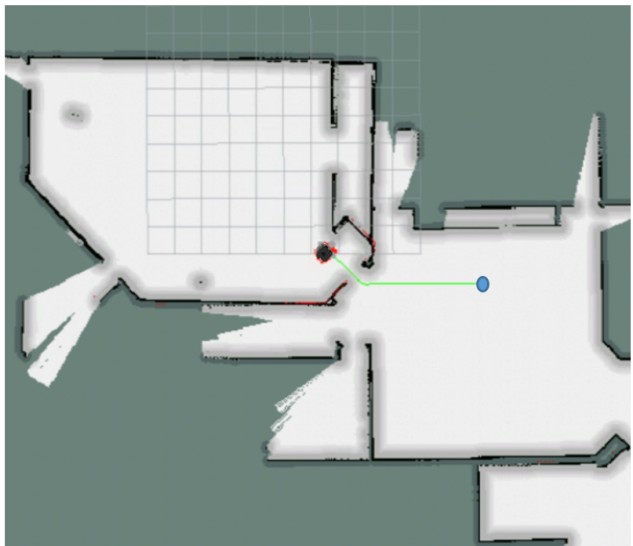

**Figure 12.** Case study of the kidnapping scenario at $t = t\_k$.

.

A ROC of the true positive rate (TPR) and false positive rate (FPR) for obtained data of classification technique are shown in Figure 13. As it can be seen from Figure 13 that the proposed QS-2DDTW performed better than the Metric-Based Detector technique in predicting the kidnapping. LR is useful for situations in which it can predict the presence or absence of a characteristic or outcome (in this case is the kidnapping event) based on a set of predictor variables.

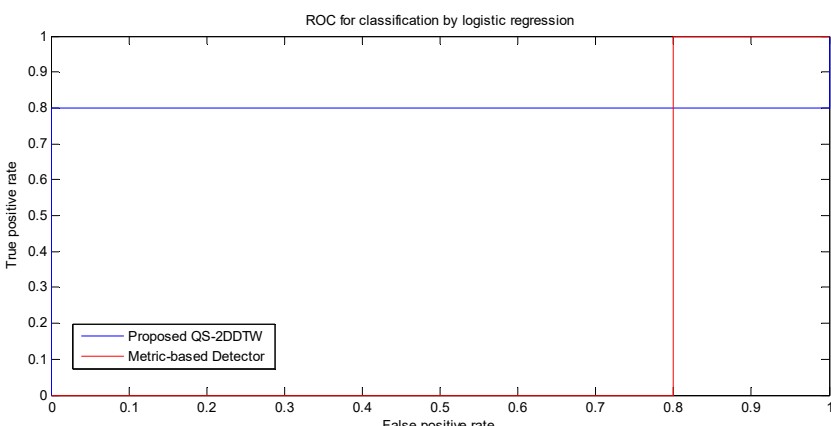

**Figure 13.** Case study of the kidnapping scenario at $t = t\_k$.

Another experimental case study for robot kidnapping is shown in Figure 14. The map shown in Figure 14 is the same as the map described previously, the differences are the trajectory of the robot and type of relocalization process. The mobile robot initially moves along the green trajectory from the start point. When it reaches the kidnapping point at $t = t\_k$, the robot is jammed in that place until it performs the relocalization process. Then it continues along the green trajectory until the desired location. This process will be

repeated few times before the performance of the detection algorithm is analyzed based on LR technique and ROC.

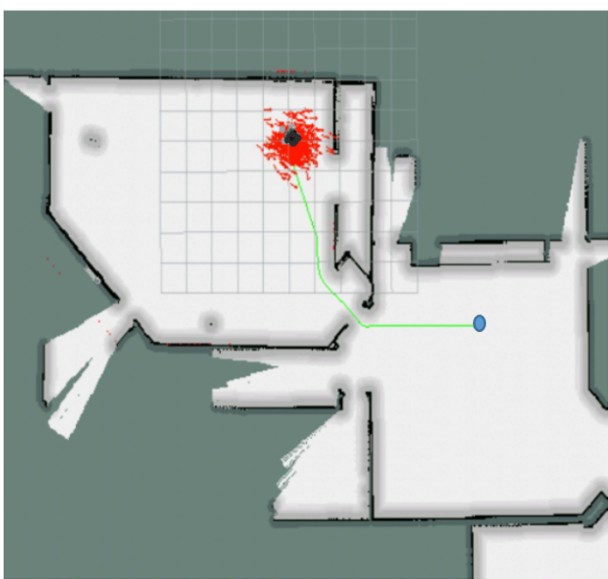

**Figure 14.** Case study of the kidnapping scenario with relocalization process.

Figure 15 shows a ROC of kidnapping scenario with relocalization process. In the experiment with QS-2DDTW, the kidnapping was successfully detected in which the proposed method successfully distinguished normal conditions around kidnapping points from the kidnapping event at the kidnapping points with low dependency of relocalization process. However, the detection performance of QS-2DDTW in the experimental case study is clearly not the same as in the simulation as nonuniform Gaussian noise distribution cannot be guaranteed to follow the Gaussian distribution in a real environment, resulting in low accuracy of SLAM and the proposed algorithm.

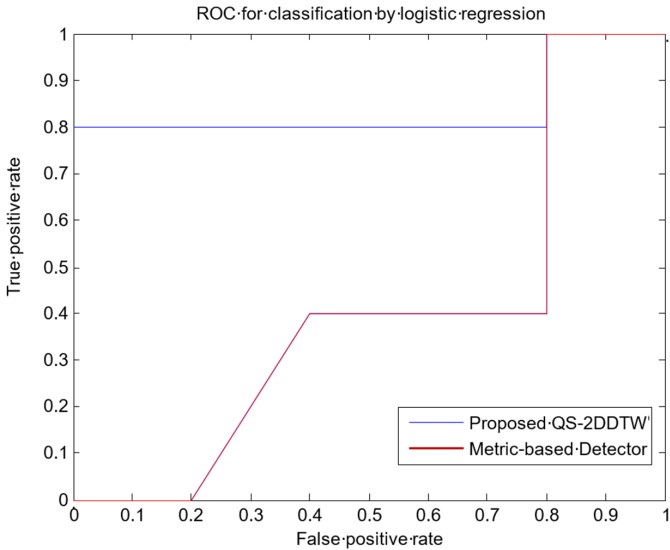

**Figure 15.** Case study of the kidnapping scenario with relocalization process.

## 6. Conclusions

An efficient method for a mobile robot to detect a kidnapping problem in range-finder-based indoor localization called Quasi Standardized 2-D Dynamic Time Warping or QS-2DDTW was presented in this paper. The proposed method specifically for mobile indoor

operation is based on the similarity degree between the two-dimensional geometry of the environment extracted from range sensor scan between two consecutive time instances. A high similarity degree indicates natural movement while the low similarity degree means there is a drastic change in the environment, which may be caused by a kidnapping event. The simulation result against MCW, ME, and the four metrics in the metric-based detector shows a high independence towards the localization process that ensures the accuracy would not drop even without the relocalization process. The performance of the proposed method, which is indicated by the ability to distinguish conditions at the kidnapping point and all other time instances around it, outperforms all of the benchmarks. In the future, this work can be extended adding more parameters related to the actual data obtained from multimodal sensors as well as evaluating other kidnapping scenarios. Moreover, it is important to improve the robustness in the kidnapping detections by taking into account the sensor fusion in a system model and the construction of the semiphysical mobile platform operated under variation of the kidnapping event within range-finder-based indoor localization.

**Author Contributions:** Z.H.I.: Conceptualization; Methodology; Validation; Investigation; Visualization; Supervision; Data Curation; Writing—original draft; Writing—Reviewing and Editing. I.B.: Formal analysis; Methodology; Validation; Investigation; Visualization; Data Curation; Writing—original draft; Writing—Reviewing and Editing. All authors have read and agreed to the published version of the manuscript.

**Funding:** This work was supported in part by the Ministry of Higher Education Malaysia and Universiti Teknologi Malaysia under Grant no. FRGS/1/2020/TK02/UTM/02/2, R.K130000.7843.5F348 and R.K130000.7643.4C355.

**Institutional Review Board Statement:** Not applicable.

**Informed Consent Statement:** Not applicable.

**Data Availability Statement:** Not applicable.

**Acknowledgments:** Thanks to UTM for providing their available software and robotic platform.

**Conflicts of Interest:** The authors declare no competing interests.

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
