# Peer review of "Efficient Detection of Robot Kidnapping in Range Finder-Based Indoor Localization Using Quasi-Standardized 2D Dynamic Time Warping"

_applsci, doi:10.3390/app11041580_

Round 1
Reviewer 1 Report
This manuscript proposes an augmented online approach to detect the kidnapping event within range-finder-based indoor localization. The authors claim that the algorithm can detect kidnapping across all-time instances of an indoor mobile robotic operation with high accuracy and maintain high accuracy in the face of re-localization failures. The approach is based on the similarity degree of geometry shape of the environment obtained from range scan data between two consecutive time instances. The authors performed a series of simulations to compare the proposed metric with the four well-known metrics.
Remark 1: Does the introduction provide sufficient background and include all relevant references?
In general, the "Introduction" or "Related work" sections should include a more detailed analysis of the presented approach. A good idea would be to analyze why the proposed measure performs better from a theoretical point of view than the others, that is, which properties of the measure make it more robust when compared to others. The particularly interesting issue for me is to clarify the simulation. If the simulation is generated by adding Gaussian noise to the perfect measurement, then the Kalman filter will remove it almost completely. As a result, the problem addressed in the article becomes trivial. Therefore, please follow the suggestions from Remark 2.
Please also refer to some research in this field like for example:
QP-DTW: Upgrading Dynamic Time Warping to handle Quasi Periodic Time Series Alignment by Imen Boulnemour and Bachir Boucheham,
Kidnapping Detection and Recognition in Previous Unknown Environment by Yang Tian and Shugen Ma
Remark 2: Is the research design appropriate?
The idea of using Dynamic Time Warping for time series comparison is well-known. This is not novel. The same as using standardization prior to calculation of DTW, see for example: "Time-synchronized clustering of gene expression trajectories" by RONG TANG and HANS-GEORG MULLER. They use standardization prior DTW. 2D representation is only the form of storing the data (when comparing two time-series, it does not influence the difference). As a result, the whole concept of Quasi-Standarized 2D Dynamic Time Warping is not a novel concept for me. The main novelty lies in applying it to the detection of robot kidnapping, which is acceptable for me for publication in Applied Science. Nevertheless, the experiment must be extended by adding more realistic scenarios. The best would be to:
- perform the experiment with the robot in the laboratory, not only the simulation,
- perform several experiments by replacing the robot in different places and different way (eg. By rotating it)
- verify how the proposed measure performs when the surrounding environment changes, such as moving objects, opening doors, etc. Won't it deteriorate the performance of the proposed measure?
- It would be beneficial to introduce some benchmarks to compare the metric, like false positive and false negative rate.
Remark 3: Are the results clearly presented?
The results are clear. However, they should be extended, see Remark 2
Remark 4: Are the conclusions supported by the results?
No. The conclusions are too general concerning the experiment which was performed. The experiment should be extended to formulate such general results.
Reviewer 2 Report
- In the test, the robot was transported to the same location. Will different locations affect the kidnapping detection?
- The proposed QS-2DDTW method was found outperforming other methods in the paper. However, no explanation was provided. More discussions may be necessary.
- In section 3, contents describing existing works can be more concise, while your method should be with more details.
Reviewer 3 Report
General remarks
This paper proposes a method to detect kidnapping conditions for a ground robot equipped with laser telemeters. Existing work around the kidnapping problem is first presented to help the reader understand the problems behind. The proposed method is an improvement of an existing one which is explained with 2 scenarios as examples, before detailing their limitations and proposing an improvement. Simulated comparisons with existing other approaches show that there are less false detections or missing ones. Additionally, it is checked whether the methods are behaving differently if the robot managed to run a relocalization successfully or not after the kidnapping, and it turns out that the new method is independent on that, contrary to other methods.
The paper is well written and describes correctly the contribution. Although this work is interesting, one may think that in practice for a robot we often have access to the outputs sent to the actuators as well as a model (the localization algorithms, not detailed in the paper, probably use them) so a contradiction between a model (e.g. state equations) only taking into account the actuators data, and a model (or your method) taking into account the sensors would probably help to be more robust in the kidnapping detections.
Typos or other remarks
L125
extra dot
L155
extra space?
L162
to find to sequences->to find sequences
L182
D^t-1and Q->missing space before and?
Table 2 and L364
See if the robot can detect that it has been kidnapped actually has been kidnapped->extra has been kidnapped?
L318
a series of simulation that include laser measurements from the Turtlebot3->it is not clear if it is real data or simulated data, from the next paragraph we understand that is simulated data?
Round 2
Reviewer 1 Report
The authors extended the research as suggested. Therefore I consider the article to be ready for publication.
Reviewer 2 Report
Satisfied with the changes